# Estimation of Respiratory Rate from Thermography Using Respiratory Likelihood Index

**DOI:** 10.3390/s21134406

**Published:** 2021-06-27

**Authors:** Yudai Takahashi, Yi Gu, Takaaki Nakada, Ryuzo Abe, Toshiya Nakaguchi

**Affiliations:** 1Department of Medical Engineering, Faculty of Engineering, Chiba University, Chiba 263-8522, Japan; kokui0820@gmail.com; 2Department of Bioengineering, School of Engineering, The University of Tokyo, Tokyo 113-8654, Japan; 3Department of Emergency and Critical Care, Graduate School of Medicine, Chiba University, Chiba 263-8522, Japan; taka.nakada@nifty.com (T.N.); ryuzoabe@chiba-u.jp (R.A.); 4Center of Frontier Medical Engineering, Chiba University, Chiba 263-8522, Japan; nakaguchi@faculty.chiba-u.jp

**Keywords:** thermal imaging, vital sign measurement, deep learning, object detection, signal processing, likelihood index

## Abstract

Respiration is a key vital sign used to monitor human health status. Monitoring respiratory rate (RR) under non-contact is particularly important for providing appropriate pre-hospital care in emergencies. We propose an RR estimation system using thermal imaging cameras, which are increasingly being used in the medical field, such as recently during the COVID-19 pandemic. By measuring temperature changes during exhalation and inhalation, we aim to track the respiration of the subject in a supine or seated position in real-time without any physical contact. The proposed method automatically selects the respiration-related regions from the detected facial regions and estimates the respiration rate. Most existing methods rely on signals from nostrils and require close-up or high-resolution images, while our method only requires the facial region to be captured. Facial region is detected using YOLO v3, an object detection model based on deep learning. The detected facial region is divided into subregions. By calculating the respiratory likelihood of each segmented region using the newly proposed index, called the Respiratory Quality Index, the respiratory region is automatically selected and the RR is estimated. An evaluation of the proposed RR estimation method was conducted on seven subjects in their early twenties, with four 15 s measurements being taken. The results showed a mean absolute error of 0.66 bpm. The proposed method can be useful as an RR estimation method.

## 1. Introduction

Respiration along with pulse, blood pressure, and body temperature is considered a critical vital sign for monitoring human health status. Observing respiratory rate (RR) and depth is important for early disease detection and physical state management [1,2,3]. Specifically, it has been reported that an RR higher than 27 breaths/min was the most important predictor of cardiac arrest in hospital wards [4], and that 21% of ward patients with an RR of 25–29 breaths/min assessed by critical care outreach service died in hospitals [5].

RR is particularly important for providing appropriate pre-hospital care during emergencies. Although different medical emergency response teams have different definitions of “abnormal” RRs, recent evidence suggests that adults with RRs higher than 20 breaths/min are probably unwell, and those with RR greater than 24 breaths/min are likely to be critically ill [6,7,8,9,10].

At present, popular respiration measurement methods require the attachment of sensors to the body, which not only causes discomfort, but also causes measurement errors due to body movements and cannot be used for trauma patients. In order to overcome these problems, non-contact respiration measurement methods have been actively developed in recent years.

In this paper, we propose an RR estimation system using thermal imaging cameras, which are increasingly being used in the medical field. By measuring temperature changes during exhalation and inhalation, we aim to track the respiration of the subject in a supine or seated position in real-time without any physical contact. An overview of the proposed method is shown in Figure 1. The proposed method automatically selects the respiration-related regions from the detected facial regions and estimates the respiration rate. Most existing methods rely on signals from nostrils and require close-up or high-resolution images, while our method only requires the facial region to be captured. Facial region is detected using YOLO v3, an object-detection model based on deep learning. The detected facial region is divided into subregions. By calculating the respiratory likelihood of each segmented region using the newly proposed index, called the Respiratory Quality Index, the respiratory-related region is automatically selected and the RR is estimated.

The further structure of this work is described as follows: Section 2 provides an overview of related works in the field of RR monitoring. Section 3 describes the proposed algorithm for RR extraction. Section 4 presents the experimental setup, face detection results, and RR estimation results. Section 5 analyzes the results of the proposed method and describes the limitations of the algorithm.

## 2. Related Works

At present, a popular respiratory tracking method involves attaching a band to the subject’s thorax [12]. A piezoelectric sensor built into the band measures the changes in the thorax caused by breathing. Another method that uses the changes in the chest is a system that measures RR during sleep using an accelerometer built into the patient’s bed [13].

An approach for estimating the respiratory cycle by measuring the temperature change of the airflow around the nose detected by a PVDF sensor attached around the nasal cavities has also been proposed [14].

These methods, however, require that the sensor is attached to a human body, e.g., to the chest or nasal cavity, which might induce stress and discomfort. The sensor may also get detached when the patient moves, resulting in erroneous measurements [15]. Furthermore, such sensors cannot be applied to patients with major trauma. As for the bed with a built-in sensor, it is not suitable for use in an ambulance, which shakes during transportation.

For these reasons, respiratory measurements in emergencies often rely on the visual and auditory senses of the person taking the measurements. Consequently, inaccuracies have been reported in some studies [16,17,18]. Although RR is considered important, it is often ignored in actual emergency situations, owing to the lack of suitable measurement techniques.

Given the aforementioned concerns, there has been considerable effort towards developing non-contact RR monitoring techniques in recent years. Marchionni et al. used a laser Doppler vibrometer, which measures vibrations of a surface, for contactless monitoring of respiration and heart rate [19]. However, this method has the disadvantage that the measurement cannot be performed when body movements other than breathing occur.

Imaging technologies for visible, near-infrared, mid-infrared, and long-infrared wavelengths have also been studied. Tan et al. proposed a system for estimating RR from chest motions recorded by a single RGB camera [20]. The method employs image and signal processing techniques to extract information on chest and abdominal movements from a sequence of video images recorded using a single video camera. For respiration measurement using an RGB camera, methods using deep learning have been actively developed in recent years [21,22]. The problem with these methods is that they are affected by the subject’s clothing and measurement is not possible when the subject moves. Infrared thermography, also known as thermal imaging, is a very promising approach for unobtrusive and non-contact monitoring of RR. In contrast to visible and near-infrared imaging systems, thermal imaging does not require any active illumination. As it is a completely passive technology that works in total darkness, it is a promising monitoring and diagnostic technique for use in a wide range of medical fields [23].

Abbs et al. proposed an algorithm for estimating RR using the continuous wavelet transform based on the temperature change around the nasal cavity obtained from thermal images [24]. In this study, the region of interest (ROI) needs to be selected manually in the first frame. Pereira et al. improved on the method of Abbs et al. by implementing a tracking algorithm [25]. In this method, the face is detected using Otsu’s binarization-based face region detection algorithm. The area around the nasal cavity is then detected based on anatomical features. Fei et al. also proposed a method to estimate RR by normalizing the respiratory temperature waveform obtained from the nasal cavity detected using the coordinated tracking algorithm and then performing wavelet transform on the resampled signal [26]. Lewis et al. proposed a method to estimate RR by tracking the nasal cavity based on the Piece-wise Bezier Volume Deformation model [27].

However, most of the non-contact monitoring methods described above require the subject to remain stationary, making it difficult to apply them to ambulances, which are subject to shaking during transport. In addition, thermal imaging cameras do not provide accurate measurements when the nose and mouth regions are not visible in the image.

In light of these concerns, we propose a system that extracts respiratory-related signals from facial regions detected with a deep learning model using a newly defined respiration likelihood index, RQI. This system enables us to measure respiration independent of the detection of nose and mouth regions.

## 3. Methodology

### 3.1. Overview of the Algorithm

The overview of the proposed system is shown in Figure 1. First, a rectangular region containing the patient’s face is detected using YOLO v3 [11], an object detection model based on deep learning. The detected face region is divided into 4 × 6 subregions. Signal intensity is extracted from each subregion, and signal processing is applied. By calculating the respiratory likelihood of each segmented region using the newly proposed index, RQI, the respiratory-related region is automatically selected and RR is estimated.

### 3.2. Detection of Facial Region

Object detection methods based on deep learning can be broadly classified into two types: two-stage and one-stage. Two-stage methods first estimate the region candidates (region proposal) and then estimate the class and bounding box coordinates. In contrast, the one-stage type estimates the class and bounding box coordinates directly without a region proposal.

Examples of a typical two-stage object detection model includes Region Based Convolutional Neural Networks(R-CNN) [28], Fast R-CNN [29], and Faster R-CNN [30]. The two-stage models are computationally expensive, and even the relatively fast Faster R-CNN has a processing speed of about 5 frames per seconds (fps) on average. To address this concern, a one-stage object detection method has been studied, which simultaneously estimates the class and bounding box coordinates.

Single Shot Multibox Detector (SSD) [31] and YOLO (You Only Look Once) [32] are typical methods for one-stage object detection.

YOLOv3 [11], which is an improved version of YOLO, achieves faster detection with the same performance as the state-of-the-art detection models (Figure 2). More specifically, YOLO v3 can detect 320 × 320 [pixel] images at about 1/3rd of the processing time while maintaining the same detection accuracy as SSD.

The respiratory monitoring of emergency patients, which is the focus in this study, requires real-time performance. For this reason, we use YOLOv3, which is capable of high accuracy and high-speed detection, to detect facial regions in thermal images.

### 3.3. Extraction of Signal Intensity

We divide the region detected by YOLOv3 into 4 × 6 regions (because the aspect ratio of a typical face is $1:\sqrt{2}$). For the *k*th ROI, the signal intensity $S_{ROI_{k}}\left(t\right)$ at timestamp *t* is calculated as the mean value within the ROI as follows:(1)$$S_{ROI_{k}}\left(t\right)=\frac{1}{wh}\sum_{i=0}^{w-1}\sum_{j=0}^{h-1}I_{ROI_{k}}\left(i,j,t\right)$$

The width and height of $ROI_{k}$ are denoted by *w* and *h* respectively, while $I_{ROI_{k}}\left(i,j,t\right)$ denotes the intensity at pixel *i*, *j* at the time stamp *t*.

Firstly, we apply a rectangular window to the signal. A rectangular window has a narrow dynamic range but a high frequency resolution.

Considering that frequency resolution is important in RR estimation, we chose the rectangular window as the window function. The length of the window was set to 15 s because of the trade-off between time resolution and frequency resolution.

Second, each signal is detrended, that is, the linear trend is removed from the signal. Detrending is expected to improve the accuracy of spectrum analysis in the low-frequency region. As will be discussed in the next section, given that the respiratory-related frequency is relatively low, we believe that trend removal is effective in respiration rate estimation.

### 3.4. Likelihood Index for Respiratory Signal

The signal intensity $S_{ROI_{k}}\left(t\right)$ was calculated for each grid from the previous section. However, the degree of relevance to respiration differs depending on the region. For example, it is expected that the area around the nose and mouth region contains more respiratory-related signals than the area containing a large amount of hair. In order to select regions that contain more respiratory-related signals, we examine the respiratory likelihood index of each grid based on the characteristics of signal intensity. The likelihood index was considered from two perspectives: RR analysis and spectrum analysis. Each of them is explained below.

#### 3.4.1. RR Analysis

Typical methods for estimating RR can be classified into two types: frequency domain analysis and time domain analysis. We consider that the higher the number of respiratory-related signals that are included, the greater the agreement between the RR estimates of both methods. For the signal intensity $S_{ROI_{k}}\left(t\right)$ in each region, the methods of estimating RR by frequency domain analysis and time domain analysis are described as follows.
(A)Estimation of RR from frequency domain:

Power spectrum density is estimated for the signal preprocessed in the previous section. A periodogram is used to estimate power spectrum density. The power spectrum density P(f) of the signal S(t) with the number of data points *N* and sampling rate $f_{s}$ is expressed as follows:(2)$$P\left(f\right)=\frac{1}{Nf_{s}}{\left|\sum_{t=0}^{Nf_{s}-1}S\left(t\right)\exp(-j\frac{2\pi t}{f_{s}})\right|}^{2}$$
P(f) is used to estimate RR. Here, we consider three different frequency bands for P(f) as follows.
(1)Low-frequency (LF) band                                 f<0.1 Hz(2)Breath-related frequency (BF) band                  0.1Hz≤f≤2Hz(3)High-frequency (HF) band                               2Hz<f

The LF region contains low-frequency noise components, and the HF region contains high-frequency noise components. The BF region is the frequency range of respiration. The BF range is defined as from 0.1 Hz to 2 Hz. The respiratory-related frequency $f_{RR_{f}}$ [Hz] is calculated as follows.
(3)$$f_{RR_{f}}=\arg\max_{0.1\leq f\leq 2}P\left(f\right)$$

As $f_{RR}$ stands for the number of breaths per second, $RR_{f}$ [bpm] is calculated as follows:(4)$$RR_{f}=60\times f_{RR}$$
(B)Estimation of RR from time domain:

To estimate RR from the time domain, we used the zero-crossing method. Zero-crossing is a method to estimate the fundamental frequency by obtaining the points where the time signal value changes from positive to negative (zero-crossing points). When *N* zero-crossing points are detected from the signal intensity $S_{ROI_{k}}\left(t\right)$, N−1 fundamental frequencies are estimated. Let the ith and i+1th zero-cross points be $t_{z_{i}}$ and $t_{z_{i+1}}$, respectively. The fundamental frequency $f_{RR_{t_{i}}}$ [Hz] is calculated as follows.
(5)$$f_{RR_{t_{i}}}=\frac{2}{t_{z_{i+1}}-t_{z_{i}}}$$

Denoting the median of $f_{RR_{t_{i}}}$ as $f_{RR_{t_{median}}}$, $RR_{t}$ [bpm] is calculated as follows:(6)$$RR_{t}=\frac{60}{f_{RR_{median}}}$$

#### 3.4.2. Spectrum Analysis

Examples of respiratory-related signal and noisy signal are shown in Figure 3a and Figure 4a, respectively. Thus, it can be seen that there are differences between the characteristics of a breath-related signal and the noisy signal (Figure 3b and Figure 4b). For each of the BF and HF ranges defined in Section 3.4.1, the characteristics of the breath-related signal are empirically considered as follows.
(A)There is only one large spectrum in the BF region.(B)There is no large spectrum in the HF region.

Based on the above characteristics, the signal likelihood index SI (Spectrum Index) in the frequency domain is defined as follows:(7)$$SI=1-\frac{1}{2}\left(F_{BF}+F_{HF}\right)$$

$F_{BF}$ and $F_{HF}$ are based on the two features of the power spectrum density P(f) in each region, representing the conditions A and B, respectively. Let the maximum spectrum in the BF region be $P_{BF_{MAX}}$ (Equation (Equation 8)). $F_{BF}$ represents the proportion of the BF region that is larger than 0.25 $P_{BF_{MAX}}$ (Equation (Equation 9)) and $F_{HF}$ represents the proportion of the HF region that is larger than 0.1 $P_{BF_{MAX}}$ (Equation (Equation 11)).
(8)$$P_{BF_{MAX}}=\max_{0.1\leq f\leq 2}P\left(f\right)$$
(9)$$F_{BF}=\frac{\int_{0.1}^{2}P\left(f\right)H_{BF}\left(f\right)df}{\int_{0.1}^{2}P\left(f\right)df}$$
(10)$$H_{BF}\left(f\right)=\left\{\begin{array}{cc} 0 & when\;P\left(f\right)<0.25P_{BF_{MAX}}\\ 1 & else \end{array}\right.$$
(11)$$F_{HF}=\frac{\int_{2}^{\infty }P\left(f\right)H_{HF}\left(f\right)df}{\int_{2}^{\infty }P\left(f\right)df}$$
(12)$$H_{HF}\left(f\right)=\left\{\begin{array}{cc} 0 & when\;P\left(f\right)<0.10P_{BF_{MAX}}\\ 1 & else \end{array}\right.$$

#### 3.4.3. Investigation of Likelihood Index

Based on the considerations in Section 3.4.1 and Section 3.4.2, we discuss the likelihood index. Figure 5 shows the relationship between the likelihood index and RR estimation error for respiratory-related signals obtained from the region where RR estimation error is lower than 1 bpm and for noisy signals obtained from the region where RR estimation error is higher than 1 bpm. $RR_{t}$ and $RR_{f}$ indicate the estimated RR obtained by time domain analysis and frequency domain analysis, respectively.

This shows that the spectrum index SI is larger and the difference in RR is smaller in the region that contains more respiratory-related signals. Based on these characteristics, we propose a new likelihood index for respiratory signals, Respiratory Quality Index. The details are described in the next section.

#### 3.4.4. Respiratory Quality Index

In this paper, we introduce the Respiratory Quality Index as a new likelihood index of respiratory-related signals. The RQI is calculated for each segmented region. $RQI_{k}$ for the *k*-th region is defined as follows:(13)$$RQI_{k}=SI\cdot G(a-|RR_{k_{t}}-RR_{k_{f}}\left|\right)$$

Based on its definition, possible values of the RQI are limited to [0, 1]. The closer the value is to 1, the more likely it is to be a respiratory-related signal.

G(x) stands for a sigmoid function. The sigmoid function is a monotonically increasing function with a value range of [0, 1] (Equation (Equation 14)). The value of *a* is a parameter, which is set to a=5 in this study because Figure 5a shows that the difference between $RR_{t}$ and $RR_{f}$ is less than 5 among most of the samples from the respiratory-related signals. The likelihood of the respiratory signal is calculated in each region by RQI. $RR_{f}$ in the region where RQI is maximum is taken as the RR of the subject.
(14)$$G\left(x\right)=\frac{1}{1+\exp(-x)}$$

## 4. Experimental Setup and Results

### 4.1. Device

The thermal imaging camera used in this study is a Boson 320 thermal imaging camera (FLIR Systems, Inc.). It has a resolution of 320 × 256 pixels, a viewing angle of 34.1° (horizontal) × 27.3° (vertical) and frames per second of 8.6. This camera captures 16-bit raw images. In the proposed system, raw images are used for signal processing, and grayscale images converted from raw images are used for image processing such as region detection.

### 4.2. YOLOv3 Training

The training data used in this study consisted of 772 thermal images captured using a Boson 320. The images obtained from the thermal imaging camera were resized to 320 × 320 pixels for training. Twelve subjects, each in the age range 20–24 years (four females and eight males) participated. The images were captured with subjects in a seated as well as a supine position (Figure 6). In the latter case, the camera was set up at a height of 1.23 m from the foot (assuming that it would be installed in the ambulance). The labeling of the facial region was done manually.

In training, fine-tuning was performed using the weights obtained by pre-training on the FREE FLIR Dataset for Algorithm Training [33].

We trained the networks (Figure 2) on the NVIDIA GTX 1080 (NVIDIA, Corp. (Santa Clara, CA, USA)). The batch size was 8 and the number of epochs was 100. We used Adam [34] as the optimizer and set the learning rate to 0.001. We trained the networks on the FLIR Dataset. In the same way, we trained using the 772 original data under the same condition.

### 4.3. Face Detection

We validate the face detector using 77 thermal images captured by Boson320. Seven healthy subjects (four females and three males) voluntarily agreed to participate in this study. Subjects were placed in the supine position. During the experiments, the camera was set up at a height of 1.23 m from the foot.

We used Intersection of Union (IoU) as the evaluation index. The IoU value between the detection box and label box was set from 0.7 to 0.8, with a step size of 0.05. The detection is regarded as successful when the IoU value is equal to or greater than the threshold value. The detection results are listed in Table 1. The results show an ${\mathrm{IOU}}_{0.70}$ of 1.00, ${\mathrm{IOU}}_{0.75}$ of 0.97, ${\mathrm{IOU}}_{0.80}$ of 0.83, and Average IOU of 0.86.

### 4.4. RR Estimation

We conducted an evaluation experiment of the proposed RR estimation method. Four 15 s measurements were performed on four female and three males subjects, 20–24 years old each, who were not included in the training data. In order to reproduce various respiratory states, the subjects were instructed to set their RR to 15, 20, 25, and 30 bpm for each measurement.

A thermal imaging camera was placed at a height of 1.23 m from the feet of a supine patient who was being photographed from above. A piezoelectric respiratory measurement belt BITalino (Plux-Wireless Biosignals, Inc., Lisboa, Portugal) was used to obtain the ground truth.

The results of RR estimation are listed in Table 2. This shows Mean Absolute Error (MAE) in the range of 0.33 bpm (patient 1) and 1.60 bpm (patient 4). Figure 7 displays a Bland–Altman plot comparing the proposed method with ground truth. This plot shows the difference of RR against the mean on the x-axis. $RR_{Pred}$ and $RR_{GT}$ stand for the RR predicted by the proposed system and ground truth, respectively. The bias average is 0.19 bpm and the 95% limits of agreement vary between −1.9 and 2.3 bpm.

## 5. Discussion

### 5.1. Face Detection

We extract facial regions in thermal images using YOLOv3, an object detection algorithm based on deep learning. The results of the experiment suggest that good agreement of ground truth and prediction was observed (IOU: 0.86, ${\mathrm{IOU}}_{0.70}$: 1.00). In the future, we will validate whether the proposed model can show good performance in actual emergency situations. In real-world scenarios, it is assumed that other people such as a clinicians will appear on the camera in addition to the patient. Therefore, we consider that it is necessary to detect the patient and the clinician as different classes, as described in [35].

### 5.2. RR Estimation

We proposed an algorithm to estimate the respiration rate from a facial region detected by YOLOv3. We introduced RQI to calculate the respiratory likelihood from the signals in a grid obtained by splitting the facial region and automatically selected the respiratory-related region using this index. The results of the experiment showed that the MAE was 0.66 bpm and the 95% limits of agreement were between −1.9 and 2.3 bpm, indicating sufficient accuracy as a respiratory monitoring method in an ambulance. Many other methods rely on the signal from the nostrils and require close-up or high-resolution images, while our method only requires the facial region to be captured. As these reported techniques based on nasal region tracking result in an MAE of 0.33 bpm [25], our method is comparable to these. One limitation of this study is the limited number of subjects for the statistical analysis. The effects of shaking assumed in the ambulance were not investigated. In addition, each of the features (e.g, $F_{BF}$, $F_{HF}$) in RQI is expected to have a different degree of association with respiratory likelihood. Although we determined the subregions to be analyzed based on the RQI in this study, we may apply machine learning methods such as SVM from these features to determine the subregions.

## 6. Conclusions

In this paper, we implemented a new non-contact respiration monitoring method based on a new introduced respiratory likelihood index. Facial region is detected using YOLOv3 [11], an object detection model based on deep learning. By calculating the respiratory likelihood of each segmented facial region using the newly proposed index RQI, the respiratory-related region is automatically selected and RR is estimated. Our method only requires the facial region to be captured while many other methods rely on the signal from the nostrils and require close-up or high-resolution images.

The results of face detection suggest that good agreement of ground truth and prediction was observed (IOU: 0.86, ${\mathrm{IOU}}_{0.70}$: 1.00). RR estimation showed that the MAE was 0.66 bpm and the 95% limits of agreement were between −1.9 and 2.3 bpm.

An RR that can be estimated from the proposed method is restricted from 6 to 120 bpm, hence it cannot measure apnea. It is very important to monitor apnea in emergencies. However, to our best knowledge, there is no system that overcomes this issue. We will investigate the changes in the distribution of RQI according to the patient’s respiratory state to identify apnea. In future work, we will evaluate this proposal method in a clinical study.

## Figures and Tables

**Figure 1 sensors-21-04406-f001:**
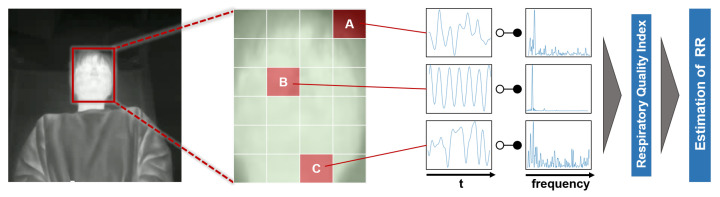
Overview of proposed method: Firstly, a rectangular region of a face is detected by YOLO v3 [11]. The detected face region is divided into 4 × 6 subregions. Signal intensity is extracted from each subregion, and signal processing is applied. By calculating the respiratory likelihood of each segmented region using the newly proposed index “RQI (Respiratory Quality Index)”, the respiratory-related region is automatically selected and RR is estimated.

**Figure 2 sensors-21-04406-f002:**
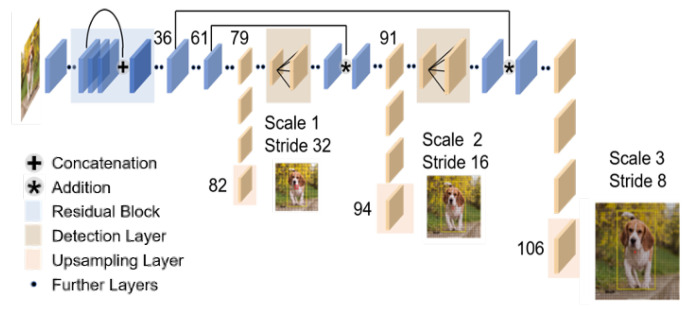
YOLOv3 architecture.

**Figure 3 sensors-21-04406-f003:**
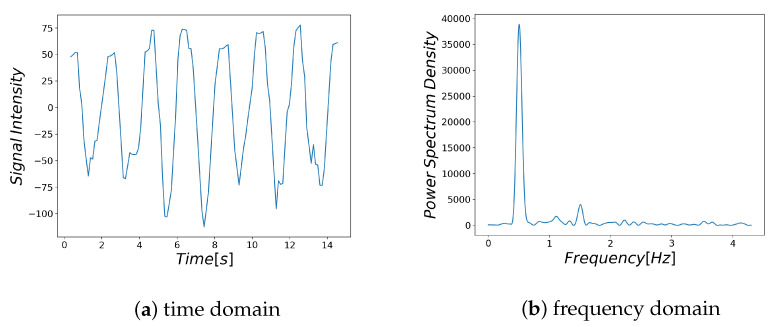
Respiratory-related signal: The signal extracted from the segmented region near the nose and its spectrum.

**Figure 4 sensors-21-04406-f004:**
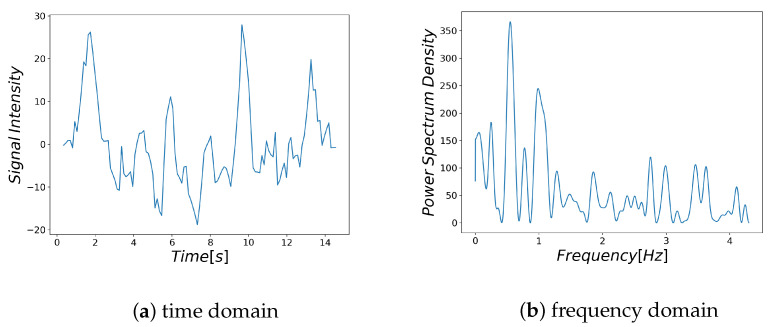
Noisy signal: The signal extracted from the segmented region far from the nose and its spectrum.

**Figure 5 sensors-21-04406-f005:**
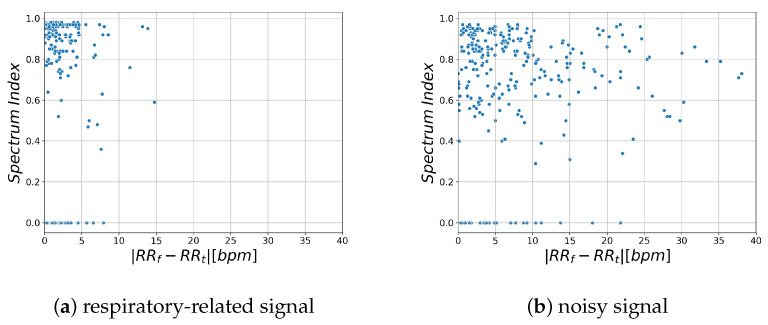
Verification of likelihood index: (**a**) shows signal obtained from the region where RR estimation error is lower than 1 bpm. (**b**) shows signal obtained from the region where RR estimation error is higher than 1 bpm. The *x*-axis represents the difference in respiratory rate [bpm] estimated from the time and frequency domains, respectively. The *y*-axis represents the Spectrum Index.

**Figure 6 sensors-21-04406-f006:**
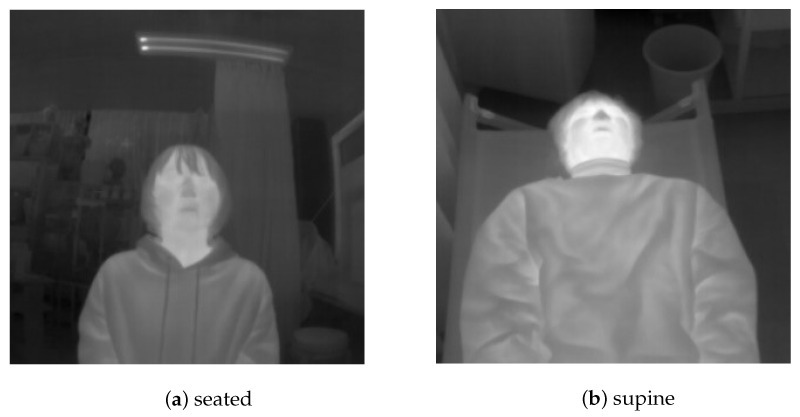
Training data: Images obtained from the thermal imaging camera for YOLOv3 training. The images were taken in a seated (**a**) and supine (**b**) position.

**Figure 7 sensors-21-04406-f007:**
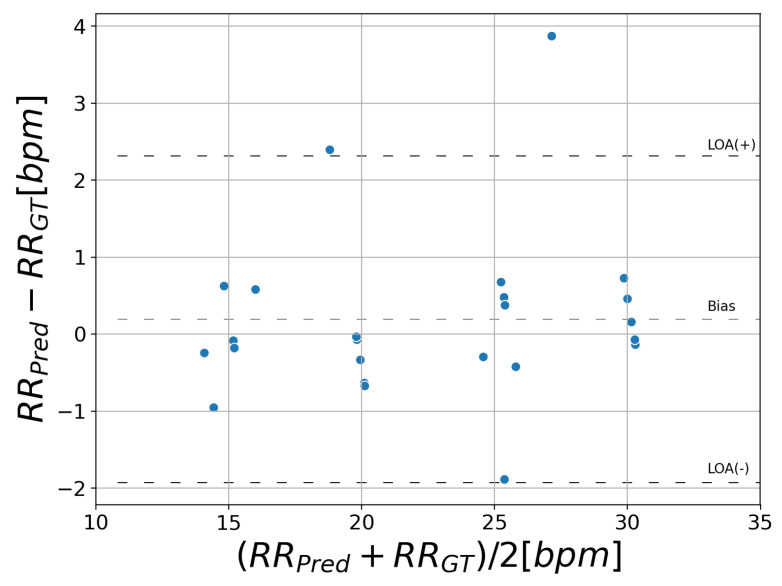
Bland–Altman plot of RR estimation: This plot shows the difference of RR against the mean on the *x*-axis. ${\mathrm{RR}}_{\mathrm{Pred}}$ and ${\mathrm{RR}}_{\mathrm{GT}}$ stand for the RR predicted by the proposed system and ground truth, respectively. The bias average is 0.19 bpm and the 95% limits of agreement vary between −1.9 and 2.3 bpm.

**Table 1 sensors-21-04406-t001:** Detection result: IoU_0.70_, IoU_0.75_, and IoU_0.80_ indicate the ratio of detection results whose IoU is higher than a certain value. For example, IoU_0.70_ shows the fraction of detection whose IoU is greater than 0.7, and Average IoU shows the Average of IoU of detection results.

Subjects	IoU_0.70_	IoU_0.75_	IoU_0.80_	Average IoU
1	1.00	0.89	0.56	0.80
2	1.00	0.90	0.30	0.79
3	1.00	1.00	1.00	0.89
4	1.00	1.00	1.00	0.89
5	1.00	1.00	1.00	0.88
6	1.00	1.00	1.00	0.91
7	1.00	0.97	0.81	0.85
Mean	1.00	0.97	0.83	0.86

**Table 2 sensors-21-04406-t002:** RR estimation result: This shows the MAE of the RR between the prediction and the ground truth.

	Subjects							Mean
	1	2	3	4	5	6	7	
MAE [bpm]	0.33	0.45	0.29	1.60	0.50	1.10	0.42	0.66

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
