# Peer review of "Estimation of Respiratory Rate from Thermography Using Respiratory Likelihood Index"

_sensors, 2021, doi:10.3390/s21134406_

Round 1
Reviewer 1 Report
Summary:
This paper proposes a respiratory estimation system based on the real-time object detection algorithm YOLOv3. By measuring temperature changes during exhalation and inhalation with thermal imaging cameras, the proposed method can automatically select the respiration-related regions from the detected facial regions and estimate the respiration rate using Respiratory Quality Index (RQI). The results of experiment demonstrate sufficient accuracy as a respiratory monitoring method in an ambulance.
Major comments:
- Overall, I find the paper is easy to follow and the experimental evaluation shows promising results.
- I would recommend the authors to provide more details on the training of YOLOv3, including other hyperparameters (e.g., learning rate and network structure) as well as the training and validation loss curves.
- Can you explain a bit more about the relationship between respiratory rate and respiratory-related frequency (Equation 4)?
Minor comments:
There is one question for clarification:
In the end, the authors indicate that an algorithm to estimate the respiratory likelihood using a machine learning model such as SVM may be able to extract respiratory-related regions more accurately.
Does it mean that simple models such as SVM could outperform YOLOv3 regarding the extraction of respiratory-related regions?
Author Response
Thank you for your sincere comments. We have carefully reviewed the comments
and have revised the manuscript accordingly. The responses according to your comment are described below point-by-point.
Hopefully, the answers and the reversed version are now suitable for acceptance and publication.
Best regards/

Reviewer 2 Report
The authors present an interesting manuscript describing the heart rate (RR) of patients from thermography using the respiratory probability index. In essence, the work is appropriate for the journal. It is well structured and describes the background and the methodology to carry out the analysis of the captured images and the likelihood index. However, the authors should consider correcting or clarifying specific points that I consider relevant to consider the work for publication:
- I think that authors should be more explicit about the aim of the study right in the Introduction. There are some places where the aim of the study is indicated, but the readers should be aware of what to expect sooner.
- Please discuss the obtained results; providing tables and figures alone with the obtained results is not enough. The discussion lacks references to other studies and a comparison of the obtained results.
- I lack the conclusion section of achieved results. I would expect to read more about consequences, i.e., how the proposed solution might be helpful. I also think in such kind of study, some space should be devoted to factors that affect the results. Could you please expand in the concluding part on these limitations of your study?
- Only twelve subjects are a limited number of participants. In this sense, it is necessary to increase the number of subjects, diversifying the age ranges to report an appropriate statistical analysis.
- Because if vector support machines (SVM) can extract respiratory-related regions more accurately, were these models not used in the developed methodology?
- The list of references should be expanded to better cover the recent debate.
Author Response

(The authors gave the same response as above.)

Reviewer 3 Report
The authors propose a new method for estimation of the respiratory rate based on deep learning and a new respiratory likelihood index.
The algorithm uses YOLO to detect the face of the patient. How did you train the CNN of YOLO to consider only faces? Have you simplified somehow the YOLO network since you are using a single class? Have you considered using a tiny version of the algorithm, since the problem is easier?
One of the problems of previous works was the necessity of the subject to remain stationary. How is your solution different since you are using a camera in a moving system? Don’t you need to guarantee stability too? Have you considered this real scenario in your tests, since it seems that your proposal is robust to this problem? You should include results proving the efficacy of your solution considering the turbulence of the ambulance.
In section 3.2, the authors refer that 772 thermal images were used to train the network. Then, they refer that FREE FLIR was used to fine tune the model. Please explain in detail the training process.
How do you explain the differences in MAE for two different patients? What is the acceptable RR in real situations?
A GPU platform is used to train the network. However, do you consider using the same platform to install in a vehicle? Maybe that it is not feasible. What are the execution times of your algorithm in a testing scenario? How long would it take in an embedded system that can be installed in an ambulance, for example?
How many images must be process a second to guarantee an analysis with enough accuracy for the tested RR? The whole testing scenario must be explained clearly and with enough accuracy, the constraints of the problem and of the solution must also be given.
The work should be compared with several previous works on the same subject using also deep learning, for example. A few previous works are:
- Bian, P. Mehta and N. Selvaraj, "Respiratory Rate Estimation using PPG: A Deep Learning Approach," 2020 42nd Annual International Conference of the IEEE Engineering in Medicine & Biology Society (EMBC), 2020, pp. 5948-5952, doi: 10.1109/EMBC44109.2020.9176231.
Shuzan, Md Nazmul Islam, et al. "A Novel Non-Invasive Estimation of Respiration Rate from Photoplethysmograph Signal Using Machine Learning Model." arXiv preprint arXiv:2102.09483 (2021).
Brieva, J.; Ponce, H.; Moya-Albor, E. A Contactless Respiratory Rate Estimation Method Using a Hermite Magnification Technique and Convolutional Neural Networks. Appl. Sci. 2020, 10, 607. https://doi.org/10.3390/app10020607
Kwasniewska, A.; Ruminski, J.; Szankin, M. Improving Accuracy of Contactless Respiratory Rate Estimation by Enhancing Thermal Sequences with Deep Neural Networks. Appl. Sci. 2019, 9, 4405. https://doi.org/10.3390/app9204405
Author Response

(The authors gave the same response as above.)

Reviewer 4 Report
General
This paper presents a well-argued case for using thermal imaging in estimating the respiratory rate of a patient that needs to be monitored for some reason. The authors develop their argument by presenting the importance of respiratory monitoring when caring for a patient under normal circumstances or being transported in an ambulance after a trauma event. The authors review methods for respiratory monitoring and outline why they might not be suitable in the situation where a patient is being transported in an ambulance. The authors then present the case for using a non-invasive method relying on thermal imaging for this situation.
The methodology for using their algorithm is presented and each step is explained clearly with reference to the appropriate literature. After using YOLOv3 for real time detection def 4 x 6 regions of interest from a patients face are used to extract the signal of interest and then using frequency and time domain analysis to determine the Respiratory Index (RR).
The paper then examines the likelihood index and its relationship to RR and presents the experimental setup along with discussions of the results.
This paper is developed in a clear, well written presentation and the results given show promise of widespread use of this technique. I look forward to seeing the results of a clinical study using this algorithm.
Comments
I was a little confused with the use of HP, BP, BF in lines 158 161. Does BP stand for Beats per Minute and if-so define it, Should HP in line 158 and 178 should be HF?
In line 183 should BP be BF?
The use of abbreviations should be defined whenever first used, as R-CNN (lines 104, 105) should be Region and Convoluted Neural Network (CNN) to allow readers not familiar with Neural Network systems a handle on what is being presented.
The investigation of using an SVM neural network (line 278) is a good track for real time processing of a face region of interest. The patient is identified and the camera pointed at the face in an ambulance. This means that there is very little distraction from the surrounds that needs to accounted for in identification of a face by a neural network in a noisy scene.
Author Response

(The authors gave the same response as above.)

Round 2
Reviewer 2 Report
I thank the authors for the changes and explanations requested in the first review of the work. This new version of the manuscript is much better than the previous one and contains the relevant elements to be published.